# Cytoskeletal Keratins Are Overexpressed in a Zebrafish Model of Idiopathic Scoliosis

**DOI:** 10.3390/genes14051058

**Published:** 2023-05-09

**Authors:** Melissa Cuevas, Elizabeth Terhune, Cambria Wethey, MkpoutoAbasi James, Rahwa Netsanet, Denisa Grofova, Anna Monley, Nancy Hadley Miller

**Affiliations:** 1Department of Orthopedics, University of Colorado Anschutz Medical Campus, Aurora, CO 80045, USAanna.monley@cuanschutz.edu (A.M.); 2Musculoskeletal Research Center, Children’s Hospital Colorado, Aurora, CO 80045, USA

**Keywords:** idiopathic scoliosis, zebrafish, RNA sequencing, intervertebral disc degeneration, transcriptomics, histology, spine, genetics, *KIF7*, keratin

## Abstract

Idiopathic scoliosis (IS) is a three-dimensional rotation of the spine >10 degrees with an unknown etiology. Our laboratory established a late-onset IS model in zebrafish (*Danio rerio*) containing a deletion in *kif7*. A total of 25% of *kif7^co^*^63/*co*63^ zebrafish develop spinal curvatures and are otherwise developmentally normal, although the molecular mechanisms underlying the scoliosis are unknown. To define transcripts associated with scoliosis in this model, we performed bulk mRNA sequencing on 6 weeks past fertilization (wpf) *kif7^co^*^63/*co*63^ zebrafish with and without scoliosis. Additionally, we sequenced *kif7^co^*^63/*co*63^*, kif7^co^*^63/*+*^, and AB zebrafish (*n* = 3 per genotype). Sequencing reads were aligned to the GRCz11 genome and FPKM values were calculated. Differences between groups were calculated for each transcript by the *t*-test. Principal component analysis showed that transcriptomes clustered by sample age and genotype. *kif7* mRNA was mildly reduced in both homozygous and heterozygous zebrafish compared to AB. Sonic hedgehog target genes were upregulated in *kif7^co^*^63/*co*63^ zebrafish over AB, but no difference was detected between scoliotic and non-scoliotic mutants. The top upregulated genes in scoliotic zebrafish were cytoskeletal keratins. Pankeratin staining of 6 wpf scoliotic and non-scoliotic *kif7^co^*^63/*c*o63^ zebrafish showed increased keratin levels within the zebrafish musculature and intervertebral disc (IVD). Keratins are major components of the embryonic notochord, and aberrant keratin expression has been associated with intervertebral disc degeneration (IVDD) in both zebrafish and humans. The role of increased keratin accumulation as a molecular mechanism associated with the onset of scoliosis warrants further study.

## 1. Introduction

Idiopathic scoliosis (IS) is a structural lateral curvature of the spine ≥10° with a rotatory component [1]. Although 2–3% of adolescents across populations are affected with IS [2,3], and the condition has been researched for decades, the etiology of IS initiation as well as the pathology of severe, progressive IS are essentially unknown. Research progress has been stymied by significant challenges, including (1) a known causal tissue, as IS may affect the skeleton [4,5,6,7], musculature [8,9,10], the proprioceptive system [11,12,13], the nervous system [14,15], joints and/or other musculoskeletal tissues [16]; (2) significant phenotypic and genetic heterogeneity across affected individuals and families [17,18,19,20]; and (3) the lack of a developmentally appropriate animal model.

Over the last decade, *D. rerio* (zebrafish) has emerged as a popular model for human scoliosis based on similar vertebral anatomy, ease of breeding, and genetic manipulation [21]. To date, multiple zebrafish genetic mutants (e.g., *ptk7* [22,23], *POC5* [24], *kif6* [25]) have linked genes involved in cilia and ciliary motility to a progressive scoliosis, giving potential insight into molecular mechanisms that may regulate scoliosis development. These mutants exhibit ciliary motility defects and often develop hydrocephalus and/or a disruption of the Reissner fiber, a fiber aggregation along the central canal that is essential to zebrafish axial development [26,27,28]. Mechanisms related to cerebrospinal fluid (CSF) flow [23,29], specialized contacting neurons within the axial sensory system (CSF-cNs) [30], inflammation [31], and urotensin peptides [32] have been hypothesized as related to these phenotypic observations; however, a definitive mechanism related specifically to scoliosis development has not been characterized.

Our zebrafish model for scoliosis, *kif7^co^*^63/*co*63^, develops juvenile-onset spinal curvatures without obvious vertebral malformations, hydrocephalus, or malformations in the Reissner fiber [33]. As patients with IS are not yet known to have gross morphological changes to these structures and develop scoliosis during the juvenile-adolescent period, *kif7^co^*^63/*co*63^ zebrafish may uniquely mirror the human condition. *KIF7* encodes a broadly conserved kinesin ciliary protein that localizes to the axonemal tip of primary cilia and binds to the plus-ends of microtubules [34]. The protein functions as a scaffold protein for ciliary function and acts as both a negative and positive regulator of the hedgehog (Hh) signaling pathway, an evolutionary conserved molecular pathway central to embryonic development, limb patterning and musculoskeletal maintenance [34], including formation and maintenance of the intervertebral disc [35]. Kif7 primarily acts by suppressing the Gli1 transcription factor. In zebrafish, Kif7 accumulates at the ciliary tip, as observed in mammals, as well as within cytoplasmic puncta, which sequester Gli1 and Gli2 and disperse in response to Hh pathway activation [36]. In humans, loss of function mutations in *KIF7* have been linked to both Joubert and the rare acrocallosal syndromes (OMIM #611254), two ciliopathies with overlapping, system-wide defects including developmental disability, skeletal abnormalities and kidney disease. Scoliosis has been observed in 5–33% of Joubert syndrome cases [37,38,39], potentially related to early hypotonia, whereas the scoliosis prevalence in acrocallosal syndrome has not yet been described.

Approximately 25% of *kif7^co^*^63/*co*63^ zebrafish develop spinal curvatures as juveniles with no evidence of abnormalities in brain morphology or hydrocephaly, and no morphological changes to the central canal cilia or the Reissner fiber. Our hypothesis is that study of the *kif7^co^*^63/*co*63^ zebrafish, both with and without the scoliosis phenotype, will provide insight into potential molecular mechanisms underlying scoliosis development. To this end we performed bulk transcriptome mRNA sequencing analyses of our mutant *kif7^co^*^63/*co*63^ embryonic (4 days post fertilization) and young adult (6 weeks past fertilization [wpf]) zebrafish with and without scoliosis, as well as age-matched heterozygous and wild-type controls. Specific findings are validated via qRT-PCR and histology.

## 2. Materials and Methods

*Kif7^co^*^63^ *generation*: *Kif7^co^*^63^ zebrafish were generated by CRISPR-Cas9 as described previously [33]. Briefly, gDNA guides were generated using the CRISPOR tool [40] and targeted to exon 3 of *kif7* (NM_001014816.1). Zebrafish embryos at the single-cell stage were injected with an injection mix created from 18.2 ng sgRNA, 50 ng Cas9 protein, 285 mM potassium chloride, 1 µL phenol red, and sterile water during the single cell phase (<45 min post fertilization). Embryos were analyzed for mutations the day after injection by lysing, PCR and fragment analysis (FA). We generated F4 homozygous *kif7^co^*^63/*co*63^ first by outcrossing CRISPR-Cas9 injected F0 to AB wild-type fish (ZIRC) to generate F1 zebrafish. F1 mutations were characterized by Sanger sequencing (see below). F1 zebrafish were crossed to AB (wild-type) zebrafish to generate F2 *kif7^co^*^63/*+*^. We then in crossed F2 *kif7^co^*^63/*+*^ to generate F3 *kif7^co^*^63/*co*63^. We then in-crossed F3 *kif7^co^*^63/*co*63^ to generate a pool of F4 *kif7*^co63/co63^. Two zebrafish indel lines were generated from mutations resulting in (1.) a 5 base pair (bp) deletion/14 bp insertion at exon 3 (RefSeq gene NM_001014816.1), Chr7:14433928 (*kif7^co^*^63^), generating a 3 amino acid insertion and (2.) a 4 bp deletion in exon 3 (RefSeq gene NM_001014816.1), Chr7:14433924 (*kif7*d4), resulting in a frameshift mutation. Mutants were outcrossed to AB wild-type fish every third generation to avoid genetic drift. Mutant *kif7^co^*^63^ was used for all experiments due to consistent percentage of scoliosis phenotype in the population [33]. A complementation test was performed using both lines to confirm the phenotype is associated with the mutation of *kif7* gene [33] (Appendix A).

### 2.1. Zebrafish Husbandry and Crossing

Zebrafish were housed at the University of Colorado Anschutz Medical Campus aquatic facility. Husbandry and experimental protocols were approved under the Institutional Animal Care and Use Committee Protocol #00370. Animals were maintained in a 14 h light/10 h dark cycle at 28.5 °C. Animals under anesthesia in 168 mg/L Tricaine were fin clipped for genotyping and RNA extraction. Animals were euthanized in 400 mg/L Tricaine, followed by immersion in ice water.

Zebrafish were bred in a sloped 1.7 L static tank. Males and females were placed in the tank separated by a divider. Each tank was set with a maximum of 8 total fish in an either 1:1 or 2:1 female to male ratio. Fish were placed in static tank in the afternoon after their last daily feeding. The following morning, water in the static tanks was changed and the dividers were pulled. Fish were monitored and embryos were harvested by siphoning after two hours.

### 2.2. Genotyping

At 24 h past fertilization (hpf), 8 embryos were euthanized by ice water immersion. Embryos were placed in 50 µL of lysis solution (500 µL 1 M Tris pH 8.3, 2.5 mL 1 M KCL, 1.5 mL 10% Tween, 1.5 mL 10% NP40, 44 mL ddH20). The 6 wpf zebrafish were genotyped by fin clip immediately post euthanasia. Briefly, zebrafish were immersed in 400 mg/L tricaine until gill movement stopped and tail fins were clipped with a sterile razor. Clipped fins were immersed in 50 µL lysis solution and run with the following reaction: 20 min at 95 °C, 2 min on ice, added 2.5 µL Proteinase K (Invitrogen, Waltham, MA, USA), 3 h at 55 °C, 10 min at 95 °C, and hold at 12 °C. Samples underwent the following PCR reaction using 12.5 µL 2× GOTaq Green Master Mix (Promega, Madison, WI, USA), 1 µL gDNA, 0.5 µL 10 µM F and R primers, and 10 µL sterile water (see Table 1):

Routine genotyping of zebrafish lines was performed by PCR as above, followed by gel electrophoresis using a 3% agarose gel with a 1:1 ratio of standard agarose to MetaPhor agarose (Lonza, Basel, Switzerland). The gel was run at 120 V for 1 h and 45 min followed by UV imaging (Appendix A).

### 2.3. RNA Extraction and Sequencing (Novogene)

At 4 days past fertilization (dpf), 20 embryos per tube were euthanized and pooled in 1 mL of RNA lysis buffer (Zymo Research, Irvine, CA, USA). Embryos were lysed using a VWR Bead Mill Homogenizer with Lysing Matrix A beads (MP Biomedicals, Santa Ana, CA, USA). Samples were iced for 5 min, lysed for 45 s, then iced again for 5 min. This process was completed three times to ensure maximal lysis. The same protocol was used to extract RNA for 6 wpf zebrafish, but 1 fish was used per tube. RNA was extracted using the QuickRNA Miniprep (Zymo) using the on-column DNase digestion protocol.

### 2.4. Bioinformatic Filtering

Bioinformatic filtering was performed by the Novogene Company (Beijing, China) following an established pipeline. Briefly, Illumina image data were base called using CASAVA and stored in FASTQ format. The sequencing error rate is represented in Qphred scores. Low-quality reads (reads where uncertain nucleotides constitute >10% of the total read, or when low-quality nucleotides [base quality < 5] constituted >50% of the read) and reads containing adapters were filtered out. Sequences were mapped to the zebrafish reference genome GRCz11/danRer11 using HISAT2 v.2.0.5. Gene expression levels and differential gene expression analysis were calculated through DESeq2 [41]. Multiple testing correction was conducted using Benjamini and Hochberg’s approach for controlling the False Discovery Rate (FDR).

RT-qPCR: Complementary DNA (cDNA) was created from RNA samples using the High-Capacity cDNA Reverse Transcription Kit with RNase inhibitor (Applied Biosystems, Waltham, MA, USA) using random primers. RT-qPCR was conducted using the same RNA as that submitted for RNA sequencing (see *RNA Extraction*). One microgram of RNA was used per reaction. Real-time quantitative PCR primer sequences are provided in Appendix A
Appendix A. Primers were analyzed using IDT’s OligoAnalyzer tool (Integrated DNA Technologies, Coralville, IA, USA) to ensure length, GC content, melting temp, hairpin tm, self-dimer and heterodimer parameters are well within range. All primers also checked using BLAT (UCSC Genome Browser, University of California Santa Cruz, Santa Cruz, CA, USA) to ensure specificity. Primer efficiency was tested for all primers to ensure genes of interest and housekeeping gene is comparable. SsoAdvanced SYBR Green Supermix (Bio-Rad, Hercules, CA, USA) was used for reactions according to the manufacturer’s instructions. An amount of 2 ng of cDNA was used per reaction. Reactions were run on a Bio-Rad CFX 96 instrument at 95 °C for 10 min followed by 40 cycles of 95 °C for 15 s and 60 °C for 1 min. Results were analyzed in accordance with the 2^−ddCt^ method [42]. Embryo samples were pooled (*n* = 20 embryos per pool) and adult samples were run as individuals. We conducted *n* = 3 biological replicates per run, with all samples in triplicate. All experiments were repeated at least twice by independent technicians.

### 2.5. Keratin Staining and Histology

Tissue staining was performed in accordance to the protocol outlined in [43]. The 6 wpf scoliotic and non-scoliotic fish were fixed with 10% neutral buffered formalin for 3 days at 4 °C. Fish were decalcified using 20% EDTA pH 8 for 10 days at room temperature using nutator. Fish were processed and embedded in paraffin using Tissue-Tek VIP 6 AI and Tissue-Tek TEC. The 6.5 μm sections were cut and dewaxed prior to staining with Mayer’s Hematoxylin, Phloxine B, Alcian Blue, and Orange G (Electron Microscopy Sciences, Hatfield, PA, USA, 26401-04, 26401-01, 26401-02, 26401-03). Slides were visualized using Zeiss Axio Scan. Z1 at 20× using brightfield.

## 3. Results

*kif7^co63/co63^* zebrafish develop isolated spinal curvatures: Approximately 25% of kif7^co63/co63^ zebrafish developed isolated spinal curvatures by 6 wpf [33] (Figure 1). These zebrafish do not show obvious morphological defects within the brain, central canal, or skeleton other than the spinal curvatures [33]. 

Transcriptomes cluster by sample age and genotype: To first determine overall transcriptomic changes occurring within scoliotic versus (vs.) non-scoliotic *kif7^co63/co63^* zebrafish, we performed bulk RNA sequencing of *kif7^co63/co63^*, *kif7^co63/+^* and wild-type (AB) zebrafish at 6 wpf, which is near the time of curvature onset (*n* = 3 individuals per phenotype). Additionally, we sequenced homozygous, heterozygous, and AB embryos (4 days past fertilization [dpf]; *n* = 3 embryo pools [20 embryos per pool] per genotype). Sequencing information is provided in Appendix A. The 6 wpf timepoint was selected for sequencing to correspond to just after the onset of spinal curvature, when scoliosis status is able to be distinguished between individuals [33]. The 4 dpf timepoint was selected due to the expression profile of *kif7* (Appendix A), which shows an increase in expression at day 5 in *kif7^co^*^63/*co*63^ embryos. At 4 dpf, embryos show a reduced expression of *kif7* (0.545 fold change, 95% CI [0.450–0.660]) and are referred to as hypomorphic.

Principal component analysis (PCA) of the bulk RNA sequencing results revealed overall clustering of transcriptomes based on zebrafish age and genotype (Figure 2). Scoliotic and non-scoliotic homozygous *kif7^co63/co63^* also clustered together on the PCA plot, suggesting that transcriptomes were globally similar when compared to other samples.

*kif7^co63/co63^* zebrafish have reduced *kif7* levels: As expected, *kif7* mRNA was reduced in 6 wpf scoliotic *kif7^co63/co63^* zebrafish compared to AB (0.5796 fold change, *p* = 5.76 × 10^−11^) (Appendix A). RT-qPCR confirmed this reduction in both 6 wpf scoliotic (0.392 fold change, 95% CI [0.314, 0.489]) and non-scoliotic zebrafish as compared to AB (0.387 fold change, 95% CI [0.247, 0.606] (Appendix A, Appendix A). No difference in *kif7* was seen between scoliotic and non-scoliotic mutant zebrafish (1.102 fold change) by RNA sequencing or RT-qPCR. *Kif7* mRNA was not detectable within 4 dpf embryos in bulk RNA sequencing; however, RT-qPCR results indicated variable kif7 expression in homozygous *kif7^co^*^63/*co*63^ embryos compared to AB embryos (0.542 fold change, 95% CI [0.142, 2.07]) (Appendix A, Appendix A).

Cytoskeletal keratins are upregulated in scoliotic *kif7*^co63/co63^ zebrafish: Differential gene expression analysis showed that, between scoliotic and non-scoliotic *kif7^co^*^63/*co*63^ zebrafish, 188 genes were downregulated, and 236 transcripts were upregulated (padj < 0.05) (Appendix A). The top upregulated genes in scoliotic vs. non-scoliotic 6 wpf zebrafish were cytoskeletal keratins, including *krt4* (174.85 fold change, *p* = 2.79 × 10^−289^) and *zgc:158846*, which is predicted to be orthologous to human *KRT8* (Table 2, Figure 3); however, this upregulation of *krt4* and *zgc:158846* was not reflected in RT-qPCR data (Figure 4). *Ces2* was the most significantly downregulated transcript (0.0494 fold change, *p* = 8.55 × 10^−114^). Gene expression analysis of upregulated transcripts showed an enrichment in 23 Gene Ontology (GO) and 16 InterPro terms, including “keratinization” (GO:0031424, *p* = 0.038) and “type II keratin” (IPR003054, 0.035). The most enriched terms among upregulated transcripts were “endoplasmic reticulum lumen” (GO:005788, *p* = 0.0018) and “somatomedin B domain” (IPR001212, *p* = 0.0039). Among downregulated transcripts, 28 GO and 17 InterPro terms were enriched, including the immune-specific GO terms “immunoglobulin complex, circulating”, “immunoglobulin receptor binding”, “positive regulation of B cell activation”, “complement activation, classical pathway”, “antigen binding”, and InterProt terms “Immunoglobulin-like fold”, “Immunoglobulin subtype”, “Immunoglobulin C1-set”, “Immunoglobulin V-set”, “Immunoglobulin-like domain”, and “Immunoglobulin I-set”.

Sonic hedgehog signaling (Shh) genes are largely unchanged in *kif7^co^*^63/*co*63^ zebrafish: Kif7 is known to function both within sonic hedgehog signaling (Shh) pathway and as an organizer of the axonemal tip of the primary cilia compartment [34]. Common transcriptional targets of hedgehog signaling include *ptch1* and *gli1* [44,45], which can serve as indicators of Shh pathway activation. We observed that *ptch1* transcripts were elevated in both 4 dpf embryos (9.816 fold change, *p* = 1.16 × 10^−5^) and 6 wpf scoliotic zebrafish (50.96 fold change, *p* = 4.02 × 10^−35^) compared to AB. *Gli1* was slightly upregulated in 6 wpf scoliotic *kif7^co^*^63/*co*63^ zebrafish compared to AB (1.461 fold change, *p* =1.05 × 10^−5^). Additional RT-qPCR data for Shh pathway genes are provided in Appendix A. Cilia genes as listed in the SYSCILIA database [46] appear to be largely unchanged in *kif7^co^*^63/*co*63^ zebrafish as compared to AB (Appendix A. Although we did not obtain usable data for Shh genes in scoliotic vs. non-scoliotic *kif7^co^*^63/*co*63^ zebrafish through RNA sequencing, RT-qPCR results suggested that there was little difference in these genes (Figure 4, Appendix A), with the exception of *dlg5a* (15.225 fold change, 95% CI [9.727, 23.830]) and *gli2* (1.574 fold change, 95% CI [1.123–2.206]).

Pankeratin staining confirms an upregulation of keratins in scoliotic *kif7^co^*^63/*co*63^ zebrafish: To confirm an enrichment of keratins in scoliotic *kif7^co^*^63/*co*63^ zebrafish compared to genotype-matched *kif7^co^*^63/*co*63^ non-scoliotic zebrafish, we performed pankeratin staining of 6 wpf *kif7^co^*^63/*co*63^ zebrafish. Scoliotic zebrafish displayed increased levels of keratins and prekeratins across muscle tissues in whole mount sections (Figure 5). Additionally, scoliotic zebrafish displayed increased keratins within the intervertebral disc (IVD).

## 4. Discussion

In this work, we build on previous findings in which human data supported by a unique zebrafish animal model suggest that mutations in *KIF7* may contribute to the IS phenotype [33]. Exploration of our unique zebrafish model, *kif7^co^*^63/*co*63^, which develops spinal curvatures during the juvenile period, supports the role of distinct molecular mechanisms which underlie the pathology of the scoliosis phenotype. Our results continue to support the role of hypomorphic mutations in *KIF7* as contributory to IS pathogenesis.

As expected, *kif7* mRNA was reduced in both scoliotic and non-scoliotic *kif7^co^*^63^ zebrafish. Overall, zebrafish transcriptomes clustered primarily by sample age and genotype. Regardless of scoliotic phenotype, homozygous *kif7^co^*^63/*co*63^ zebrafish clustered together on the PCA plot (Figure 1), indicating that the transcriptomes of these zebrafish were overall similar.

Kif7 as a ciliary kinesin has dual roles within the cell; one, in the sonic hedgehog (Shh) signaling pathway and, second, as a ciliary regulator [34]. *Kif7* is broadly expressed in human [47] and zebrafish [48]. In mouse, Kif7 functions as both a negative and positive regulator of the Shh pathway within the primary cilium downstream of Smoothened (Smo) and upstream of Gli transcription factors [49]. *Ptch1*, which encodes Patched, the primary receptor for Shh, is noted to be upregulated in 4 day (9.816 fold change, *p* = 1.16 × 10^−05^) and 6 wpf scoliotic *kif7^co^*^63/*co*63^ zebrafish (50.96 fold change, *p* = 4.02 × 10^−35^) as compared to age-matched AB zebrafish. Although we did not obtain usable bulk RNAseq data for Shh genes in scoliotic vs. non-scoliotic *kif7^co^*^63/*co*63^ zebrafish, RT-qPCR data showed little difference in Shh genes in 6 wpf scoliotic vs. non-scoliotic *kif7^co^*^63/*co*63^ zebrafish, with the exception of *dlg5a* (15.225-fold change, 95% CI [9.727, 23.830]). Dlg5a binds directly to Kif7 [50], and was observed as upregulated in *kif7^co^*^63/*co*63^ embryos over controls within our previous study [33]. Dlg5 is also a regulator of Gli1 protein ubiquitination and degradation and, most recently, is recognized as a major factor of Shh signaling, particularly in relation to glioblastoma tumors, reflecting its known role in critical processes involving cell–cell adhesion during neural development [51,52]. Although we cannot rule out the possibility that Shh signaling plays a tissue-specific role in spine morphogenesis in *kif7^co^*^63/*co*63^ zebrafish, as Shh target genes *ptch1* and *gli1-2* were largely unchanged between scoliotic and non-scoliotic mutants, the upregulation of *dlg5* could represent alternative functional mechanisms of cellular interactions.

We observed striking differences in several other transcripts between scoliotic and non-scoliotic *kif7^co^*^63/*co*63^ zebrafish, notably a large upregulation in the zebrafish cytoskeletal keratins *krt4* and *zgc:158846* (*krtt2c6*), which are orthologous to human *KRT8* (fold change= 174.85, *p* = 2.79 × 10^−289^) and *KRT7*, −8 and −9 (fold change = 107.63, *p* = 1.37 × 10^−159^). This upregulation of zebrafish keratin *krt4* was too variable between specimens to draw a definitive conclusion by RT-qPCR, which may have been due to a difference in the methodologies used [53]. However, we also observed increased levels of keratins and prekeratin proteins in the dermis, musculature, and intervertebral disc (IVD) of scoliotic *kif7^co^*^63/*co*63^ zebrafish as compared to non-scoliotic genotype- and age-matched mutant zebrafish. These key transcriptomic differences may arise spontaneously and contribute to the development of spinal curvatures in scoliotic mutants, or they may be a secondary effect that develops after the initiation of spinal curvatures.

It is unknown specifically how *kif7* mutations may lead to increased expression of keratins, a known product of keratinocytes, in hypomorphic *kif7^co^*^63/*co*63^ zebrafish. Due to the role of Shh signaling in the pathogenesis of basal cell carcinoma of the skin, the role of Kif7 has been studied in keratinocytes. This work has been primarily performed in the mouse as the model system; however, studies within the zebrafish have shown similarities of epidermal ontogeny [54,55]. In keratinocytes, Kif7 has been shown to have a crucial role as a positive regulator of Shh signaling through its regulatory function of the Gli transcription factors [56]. In embryonic keratinocytes, inactivation of both *Kif7* and *Sufu*, an additional regulator of Gli transcription factors, leads to a loss of epidermal differentiation and follicular fate, while in the adult this loss leads to the induction of basal cell carcinoma [57]. As seen in histological sectioning (Figure 4), our hypomorphic *kif7^co^*^63/*co*63^ zebrafish with scoliosis exhibited significant upregulation of keratin expression overall. The identification of the specific mechanism as to what drives this strong expression of keratin within the muscle and epithelium of the hypomorphic scoliotic *kif7^co^*^63/*co*63^ zebrafish remains to be determined.

As stated, the specific human orthologues of the top differentially expressed genes within our expression data are *KRT7* and *KRT8. KRT7* and *KRT8* encode keratin-7 and keratin-8, respectively, and are members of the type II keratin family that are broadly expressed and form intermediate filaments in the cytoplasm of epithelial cells. *KRT7* expression has a role in epithelial–mesenchymal transition and its dysregulation has been highly associated with various types of cancers and tumor progression [58,59,60]. Cytokeratins of the intermediate filament subgroup, which includes KRT8, have been observed in adult mouse skeletal muscle [61]. Keratin-8, in addition to keratin-19, links the contractile apparatus of striated muscle to dystrophin [62]. More recently, *KRT8*, *KRT18* and *KRT19*, all recognized markers of notochordal cells, have been found to be expressed in a subpopulation of adult nucleus pulposus (NP) cells. NP cell degeneration is associated with intervertebral disc degeneration (IVDD), a prominent health problem worldwide [63,64]. The finding of these cellular markers within the NP has allowed for study of the intervertebral disc (IVD) and its degeneration in both zebrafish and humans [65]. The role of keratins within the IVD as a potential molecular mechanism associated with the onset of scoliosis warrants further study. Future explorations will focus on isolating the tissue or cell type driving this expression pattern.

*Ces2* and *TTC38* were significantly downregulated within the scoliotic homozygous *kif7^co^*^63/*co*63^ zebrafish when compared to non-scoliotic homozygous *kif7^co^*^63/*co*63^ fish. Both genes are related to metabolic abnormalities through enzymatic reactions of various drugs and endogenous compounds (via NCBI RefSeq [66]), to which there is no known relationship to axial skeletal development.

Collectively, we present transcriptomic data that associate an upregulation of cytokeratins with the phenotype of scoliosis within a *kif7^co^*^63/*co*63^ zebrafish animal model. This comparative transcriptomics analysis has significance in the determination of unique differentially expressed genes as related to the scoliosis phenotype. This initial step, when followed with gene-set-enrichment analyses and orthology mapping, has the potential to reveal significant associations with biological relevance to the observed phenotype of idiopathic scoliosis.

## Figures and Tables

**Figure 1 genes-14-01058-f001:**
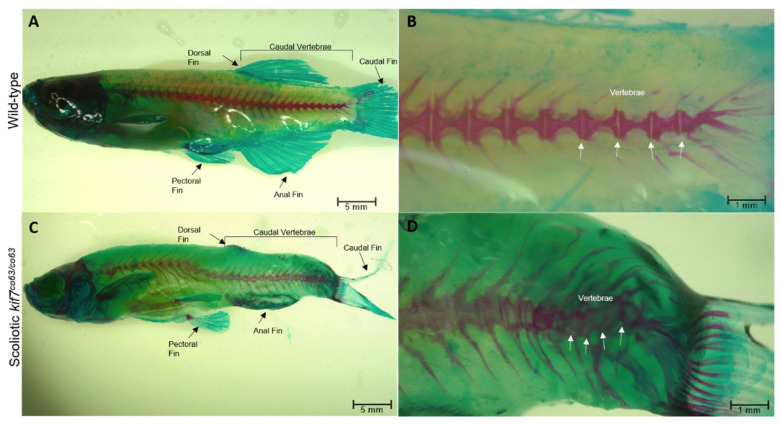
Approximately 25% of *kif7^co63/co63^* zebrafish (*D. rerio*) show spinal curvatures, as indicated. The remaining zebrafish appear identical to wild type. Spinal curvature can be seen on the caudal part of the fish compared to wild type (**A**,**C**). Clear segmentation of vertebrae was observed in both wild-type (**B**) and scoliotic fish (**D**) as denoted by arrows.

**Figure 2 genes-14-01058-f002:**
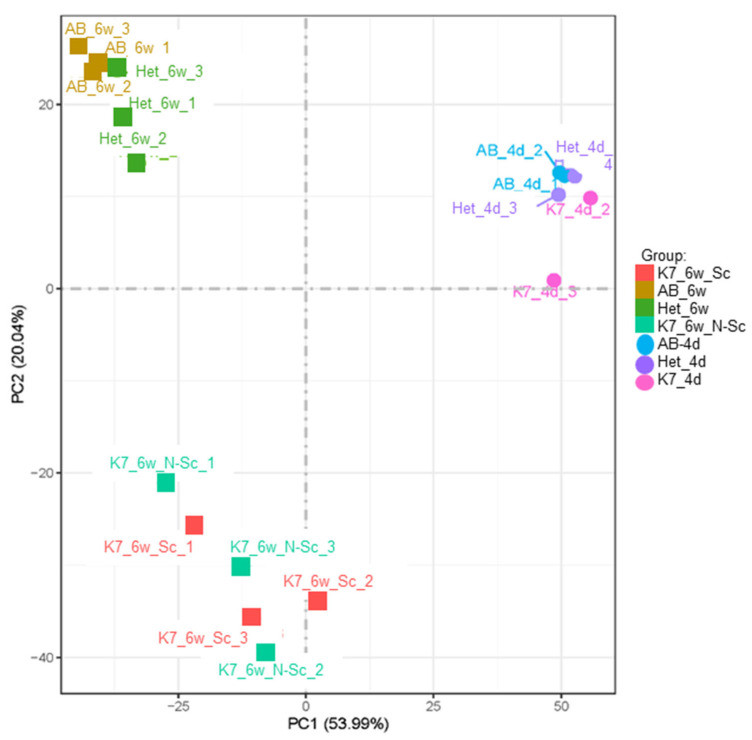
Principal Component Analysis (PCA) plot of all sequenced samples. 6 wpf (6w) *kif7^co^*^63/*co*63^ scoliotic (K7_6w_S), non-scoliotic (K7_6w_N-Sc), heterozygous (Het_6w) and AB (AB_6w) zebrafish (*D. rerio*) are shown, along with AB, heterozygous, and *kif7^co^*^63/*co*63^ embryonic zebrafish (4 dpf [4d]).

**Figure 3 genes-14-01058-f003:**
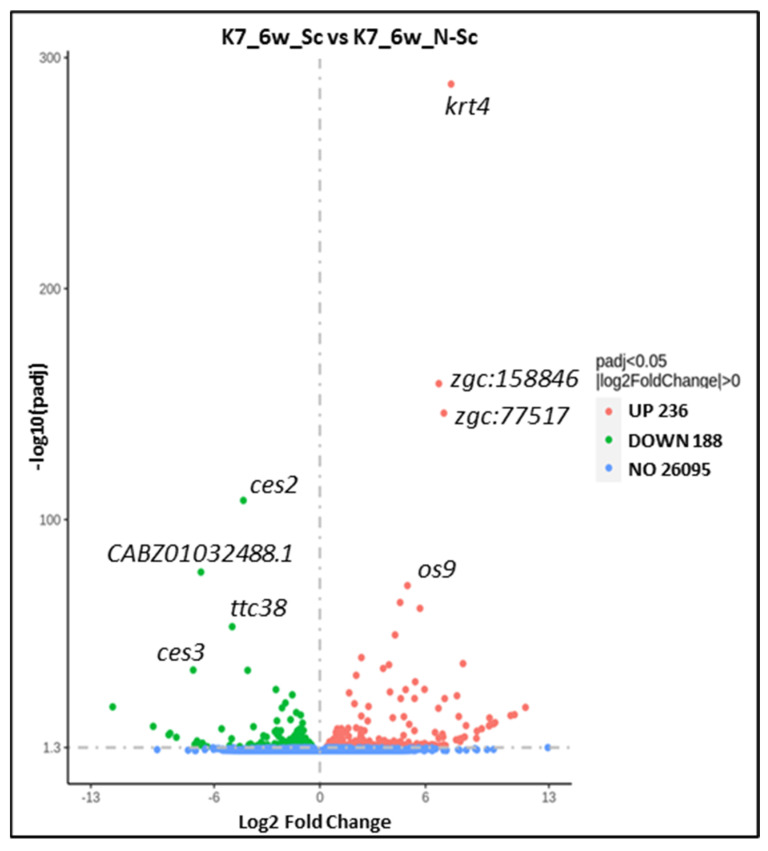
Volcano plot of 6 wpf *kif7^co^*^63/*co*63^ scoliotic (S) vs. non-scoliotic (N-Sc) zebrafish (*D. rerio*). Each point represents a single transcript, and gene names are provided for the top four up- and downregulated transcripts.

**Figure 4 genes-14-01058-f004:**
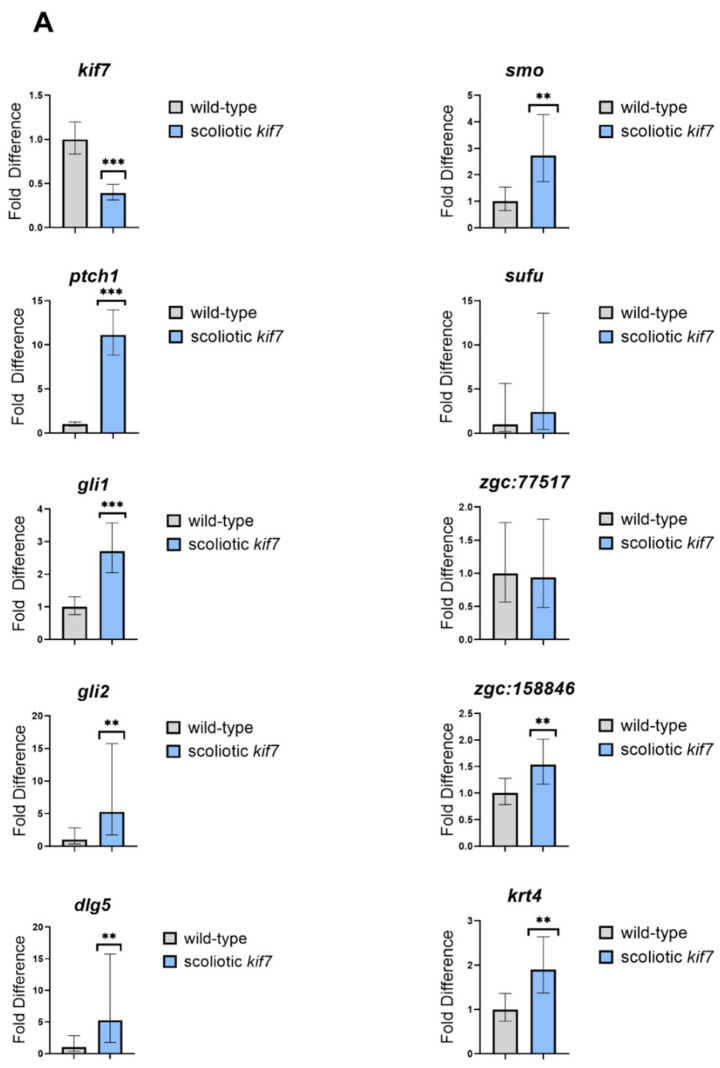
RT-qPCR of 6wpf zebrafish (*D. rerio*). Comparisons between wild-type and scoliotic *kif7^co63/co63^* (**A**), wild-type and non-scoliotic *kif7^co^*^63/*co*63^ (**B**), and scoliotic *kif7^co^*^63/*co*63^ and non-scoliotic *kif7^co^*^63/*co*63^ (**C**) was performed for genes in the shh signaling pathway and the keratin genes observed in the RNA sequencing data. Error bars represent 95% confidence intervals. * indicates *p* < 0.05, ** indicates *p* < 0.01, and *** indicates *p* < 0.001.

**Figure 5 genes-14-01058-f005:**
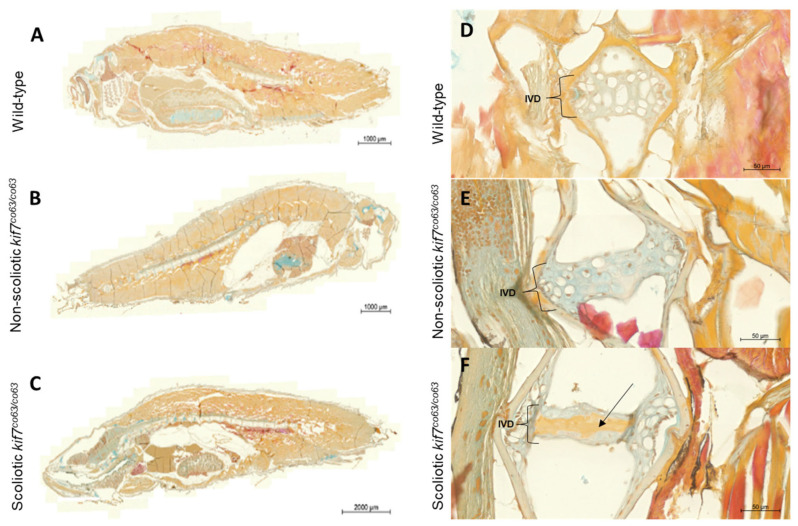
Keratin staining of wild-type (**A**,**D**), non-scoliotic (**B**,**E**), and scoliotic *kif7^co63/co^*^63^ (**C**,**F**) 6 wpf zebrafish (*D. rerio*). Keratin staining of whole cross section fish (**A**–**C**) and selective caudal vertebrae (**D**–**F**) depicting the intervertebral disk (IVD) are shown. The IVD is distinguished by vacuolated cells clearly in the wild-type and non-scoliotic *kif7^co^*^63/*co*63^ fish, but vacuolated cells are less visible in scoliotic *kif7^co^*^63/*co*63^. Blue: glycosaminoglycans; orange to red: prekeratin to keratin; brown: nuclei. Increased keratin/prekeratin can be seen in the IVD of scoliotic fish compared to non-scoliotic fish and wild type, as denoted by the black arrow. *N* = 3 zebrafish were analyzed for each group (wild type, scoliotic *kif7^co63/co^*^63^, and non-scoliotic *kif7^co^*^63/*co*63^).

**Table 1 genes-14-01058-t001:** Specifications of the PCR reaction for genotyping eight embryonic zebrafish at 24 h past fertilization (hpf).

	Initial	Denature	Anneal	Extend	Final
Temp	95	95	58	72	72
Time	3 min	30 s	30 s	30 s	20 min
		34×			

**Table 2 genes-14-01058-t002:** Table of top 10 downregulated and top 10 upregulated transcripts (by *p* value) for genetically identical scoliotic vs. non-scoliotic 6 wpf *kif7^co^*^63/*co*63^ zebrafish (*D. rerio*). Corresponding zebrafish genes and predicted human gene orthologs are provided (via ZFIN). *p* adjusted indicates values after multiple testing correction.

Fold Change	Log2 Fold Change	*p* Value	*p* Adjusted	Gene Name	Predicted Human Ortholog
0.0494	−4.34	8.55 × 10^−113^	5.20 × 10^−109^	*ces2*	*CES2*
0.0093	−6.75	1.12 × 10^−81^	5.44 × 10^−78^	*CABZ01032488.1*	*Unknown*
0.0317	−4.98	8.66 × 10^−58^	2.34 × 10^−54^	*ttc38*	*TTC38*
0.0068	−7.19	8.69 × 10^−39^	1.41 × 10^−35^	*ces3*	*CES3*
0.0583	−4.10	1.23 × 10^−38^	1.88 × 10^−35^	*si:dkey-167k11.5*	*CCDC134*
0.1780	−2.49	3.19 × 10^−30^	3.89 × 10^−27^	*pip4p1a*	*PIP4P1*
0.3392	−1.56	7.42 × 10^−28^	7.53 × 10^−25^	*ctsd*	*CTSD*
0.2588	−1.95	2.89 × 10^−24^	2.43 × 10^−21^	*si:ch211-195h23.3*	*CARD8*
0.0003	−11.76	1.63 × 10^−22^	1.24 × 10^−19^	*si:dkeyp-73d8.9*	*Unknown*
0.2253	−2.15	4.62 × 10^−22^	3.31 × 10^−19^	*lig1*	*LIG1*
174.85	+7.45	1.14 × 10^−293^	2.79 × 10^−289^	*krt4*	*KRT8*
107.63	+6.75	1.13 × 10^−163^	1.37 × 10^−159^	*zgc:158846 (krtt2c6)*	*KRT7,8,9*
131.60	+7.04	1.00 × 10^−150^	8.13 × 10^−147^	*zgc:77517 (krtt1c6)*	*KRT18*
31.34	+4.97	9.44 × 10^−76^	3.83 × 10^−72^	*os9*	*OS9*
23.59	+4.56	2.27 × 10^−68^	7.90 × 10^−65^	*si:ch73-308m11.1*	*GIMAP8*
51.27	+5.68	9.23 × 10^−66^	2.81 × 10^−62^	*si:ch211-11p18.6*	*Unknown*
19.29	+4.27	3.63 × 10^−54^	8.84 × 10^−51^	*CR774195.1*	*Unknown*
5.13	+2.36	2.49 × 10^−44^	5.52 × 10^−41^	*tmem97*	*TMEM97*
280.14	+8.13	1.12 × 10^−41^	2.27 × 10^−38^	*myl6*	*MYL6*
15.14	+3.92	4.14 × 10^−41^	7.77 × 10^−38^	*BX322618.1*	*Unknown*

## Data Availability

The data necessary for confirming the conclusions of this article are represented fully within the article and its Appendix A, including the complete lists of differentially expressed genes for all RNA sequencing samples (Appendix A). Raw data files are available through NCBI Gene Expression Omnibus (GEO), accession number: GSE231606.

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
