# Peer review of "Cytoskeletal Keratins Are Overexpressed in a Zebrafish Model of Idiopathic Scoliosis"

_genes, 2023, doi:10.3390/genes14051058_

Round 1

Reviewer 1 Report

The manuscript from Cuevas et al is the follow-up of a former one (Human Mutation. 2021;42:392–407). In this previous paper the authors associate mutations in KIF7 gene to human scoliosis and generate a model in zebrafish knocking out the orthologous gene. In the zebrafish model they describe that kif-/-zebrafish displayed severe scoliosis and alterations in Hh pathway gene expression. In this new manuscript they present bulk mRNA sequencing data comparing AB zebrafish and mutant zebrafish, and mutant zebrafish with and without phenotype.

Scoliosis is a common spinal deformity that significantly disrupts the wellbeing of patients. Its etiology and pathogenesis remain unclear, probably because of the genetic heterogeneity of scoliosis and the lack of appropriate model systems. So, the development of a zebrafish model is definitely of importance. However, the manuscript is mainly descriptive.

The manuscript needs improving.

11-      A kif7 expression profile along zebrafish development is essential. How did the authors select the stages for the rna-seq analysis? An in situ hybridization analysis could also help.

22-      Fig. 1 needs to show a normal specimen too. Also the scale bar is missing. The arrow should be explained in the legend. And also, other, major structures of the specimen should be indicated as references.

33-      Table 2 could be in the suppl. Material or at least, the formatting should be better.

4-      Fig. 2 should be enhanced. The naming of the experimental samples should be improved in order to make it easier to the reader following the results. For example, 4 dpf samples could be squares and 6 weeks pf samples could be triangles. H_d4? Also, Scoliotic could be (Sc) and non-scoliotic (N-Sc). The legend should not include conclusions...” loosely cluster together,”.

5-      Table 3. Please include top 10 up and 10 down-regulated.

6-      Fig. 3. Only two genes are identificable? os9? Ttc39? Other? Larger font would be better too.

7-      Fig 4. Please include wild type images. Also, indicate the positive staining with arrows, arrowheads, etc. for each staining. Indicate anatomical structures to help reader. Include scale bar.

8-      RT-qPCR results should be in a figure as in Human Mutation. 2021;42:392–407. Please make clear which results are from the RNA seq experiments and are confirmed by qPCR.

9-      Paragraph needs citation (lines 312-315)

10-  Revise Discussion section taking into account qPCR confirmed expression results. Also, quantification of keratin staining in notochord could be informative for discussion.

Author Response

Reviewer 1

  1. A kif7 expression profile along zebrafish development is essential. How did the authors select the stages for the rna-seq analysis? An in situ hybridization analysis could also help.

Thank you for this excellent suggestion. We had previously collected the kif7 expression profile and have also added this figure to the Supplemental File. We have added detail regarding the selection of the 4 dpf and 6 wpf timepoints within the manuscript Results section (lines 171-175). We wholly agree that an in-situ hybridization analysis would be useful, and while we completed this assay, we obtained significant off-target effects from our RNA probes, despite significant efforts. As we were unable to draw conclusions from these images, we opted to not include them with this manuscript.

  1. 1 needs to show a normal specimen too. Also the scale bar is missing. The arrow should be explained in the legend. And also, other, major structures of the specimen should be indicated as references.

Thank you for these suggestions. We have revised Figure 1 to include a wild-type zebrafish, scale bars, and labels for the major anatomical structures.

  1. Table 2 could be in the suppl. Material or at least, the formatting should be better.

We agree with this suggestion, and we have moved Table 2 to the Supplemental File along with the other sequencing QC figures.

  1. 2 should be enhanced. The naming of the experimental samples should be improved in order to make it easier to the reader following the results. For example, 4 dpf samples could be squares and 6 weeks pf samples could be triangles. H_d4? Also, Scoliotic could be (Sc) and non-scoliotic (N-Sc). The legend should not include conclusions...” loosely cluster together,”.

Thank you for pointing this out and recommending ways to improve the figure. We have enhanced Figure 2 by changing the 6 wpf samples to squares, enlarging the font, and changing the names of the scoliotic and non-scoliotic samples to S and N-Sc, as suggested. We also changed H_d4 and H_6w to het_d4 and het_6w, and we have specified in the figure legend that those are heterozygous samples. The conclusion statement in the figure legend (“loosely cluster together”) was also removed.

  1. Table 3. Please include top 10 up and 10 down-regulated.

Table 3 was changed to include the top 10 up- and top 10 down-regulated, as suggested. Please note that Table 3 is now Table 2, as the previous Table 2 was moved to the Supplemental File.

  1. 3. Only two genes are identificable? os9? Ttc39? Other? Larger font would be better too.

We have enlarged the font for the axes labels, legend, and sample names of Figure 3. We have also added additional gene labels so that the top 4 up- and down-regulated genes are identifiable, including os9 and ttc38.

  1. Fig 4. Please include wild type images. Also, indicate the positive staining with arrows, arrowheads, etc. for each staining. Indicate anatomical structures to help reader. Include scale bar.

As suggested, we have enhanced Figure 4 with a wild-type image, as well as arrows and brackets to identify the IVD and the relevant keratin staining in scoliotic kif7co63/co63. We have also added in the missing scale bars.

  1. RT-qPCR results should be in a figure as in Human Mutation. 2021;42:392–407. Please make clear which results are from the RNA seq experiments and are confirmed by qPCR.

We have added figures for our RT-qPCR results (Figure 5) within the main text and have also kept the tables of RT-qPCR results in the Supplemental File. We have added additional detail regarding the RNA-sequencing vs RT-qPCR results in the Discussion within the 3rd and 4th paragraphs (lines 278-308).

  1. Paragraph needs citation (lines 312-315)

We have added the missing citation (now on line 344).

  1. Revise Discussion section taking into account qPCR confirmed expression results. Also, quantification of keratin staining in notochord could be informative for discussion.

We have revised the Discussion section to better reflect the qPCR confirmed expression results (lines 278-308). While we agree that quantification of the keratin staining would be informative, unfortunately we are unable to quantify this type of stain with our current histological equipment. For further studies of this zebrafish model, we plan to investigate the development of a fluorescent antibody that would allow for quantification.

Reviewer 2 Report

In this current study, the authors generated a mutated allele of kif7 in zebrafish, showing spinal curvatures at 6wpf (25% of the homozygotes). The following RNA-seq results revealed that keratins and genes involved in the Shh pathway were upregulated in kif7 homozygotes. Overall, the manuscript is hard to follow without the legend of supplemental files. In addition, supplementary file 2 needs to be included. Major comments are provided below to help to improve the quality of the manuscript:

1.     The authors showed a correlation between spinal curvatures and the kif7 mRNA level reduction. However, strong evidence is required to demonstrate that the cause of spinal curvatures is due to the loss of kif7 expression, other than loss-of-target effects.  

2.     Tow mutated alleles (-5,+14, and -4) were reported in the M&M. However, it needs to be clear which one was used for the RNA seq assay. Do two alleles produce the same phenotype, assuming that the -5,+14 allele would not lead to loss of RNA expression?

3.     The agarose gel images were encouraged to be provided to show the ability to separate the wt and mutant bands, given that there are only a few bp differences. 

4.     The authors were encouraged to present the qPCR results to the main text. However, some data of less importance can be given as supplemental data, e.g., table 2 and figure 2. 

5.     Scale bars are missing from the figures. More labeling is needed in Fig.4. 

Author Response

Reviewer 2

  1. The authors showed a correlation between spinal curvatures and the kif7 mRNA level reduction. However, strong evidence is required to demonstrate that the cause of spinal curvatures is due to the loss of kif7 expression, other than loss-of-target effects.

Thank you for pointing out that we had not sufficiently described our kif7 model. The establishment of this model was primarily done in our previous publication (Terhune et al., 2020), and we have added an additional citation and description of this model within the Methods section. To assess the possibility of off-target effects, we had completed a complementation test using our different kif7 lines. We have added a brief description of this test in the Methods (lines 96-98), and a figure of the complementation test results is provided in the Supplemental Material.

  1. Tow mutated alleles (-5,+14, and -4) were reported in the M&M. However, it needs to be clear which one was used for the RNA seq assay. Do two alleles produce the same phenotype, assuming that the -5,+14 allele would not lead to loss of RNA expression?

Thank you for informing us that we were not clear on the mutation under study in this manuscript. Kif7co63 refers to the -5/+14 allele (lines 91-93). We have added a brief explanation for why Kif7co63 was used, which is that this allele most consistently produced the scoliosis phenotype (lines 95-96). This allele is the one described within our previous publication (Terhune et al., 2020).

  1. The agarose gel images were encouraged to be provided to show the ability to separate the wt and mutant bands, given that there are only a few bp differences.

Thank you for this suggestion. We have added an agarose gel image to the Supplemental File to show the separation of mutant and wild-type bands, in addition to relevant controls.

  1. The authors were encouraged to present the qPCR results to the main text. However, some data of less importance can be given as supplemental data, e.g., table 2 and figure 2.

Thank you for this suggestion. We have added qPCR results to the main text (Figure 5) and have moved Table 2 to the Supplemental File. Although we agree that Figure 2 is less important than the other figures, we have decided to keep it within the main text and enhance the image to ensure readability, as we describe this figure within the Results and Discussion. Figure 2 was enhanced by improved labeling and enlarging the font, as suggested by Reviewer 1.

  1. Scale bars are missing from the figures. More labeling is needed in Fig.4.

Thank you for pointing out this omission. We have added scale bars and anatomical labeling, as well as wild-type images, to Figure 4 as well as Figure 1.

Round 2

Reviewer 1 Report

The manuscript of Cuevas et al. has been improved. However, a few details should be corrected/clarified.

1-line181, "RT-qPCR sequencing"?

2-lines 171-174 explain the reason why 4dpf embryos were chosen for the RNA seq analysis. A kif7 expression time course along embryo development would have been better. However, the sole 4dpf expression data could work if a better explanation of the mutant is presented. The kif7co63/co63 overexpresses a mutant protein at 4 dpf? Is it truly an hypomorph?

3-the resolution of Fig 5 should be improved.

Reviewer 2 Report

I couldn’t find the updated supplementary files in the revised version of the manuscript. The authors claimed, “a figure of the complementation test results is provided in the Supplemental Material” (response to my Q1); “We have added an agarose gel image to the Supplemental File” (response to my Q3). However, the supplementary files were not updated. 

The images of Figures 1A, B, F in the revised version have been published in their previous study (figure 2, Terhune et al., 2020). How can the authors convince the reader that the same alizarin red/alcian blue staining image of the tail (figure 1F) comes from different fish? And the same wt fish shows different tail staining (figure 1C)?

The resolution of Figure 5 is too low.
